# MiR-125b-5p Is Involved in Sorafenib Resistance through Ataxin-1-Mediated Epithelial-Mesenchymal Transition in Hepatocellular Carcinoma

**DOI:** 10.3390/cancers13194917

**Published:** 2021-09-30

**Authors:** Akihiro Hirao, Yasushi Sato, Hironori Tanaka, Kensei Nishida, Tetsu Tomonari, Misato Hirata, Masahiro Bando, Yoshifumi Kida, Takahiro Tanaka, Tomoyuki Kawaguchi, Hironori Wada, Tatsuya Taniguchi, Koichi Okamoto, Hiroshi Miyamoto, Naoki Muguruma, Toshihito Tanahashi, Tetsuji Takayama

**Affiliations:** 1Department of Gastroenterology and Oncology, Tokushima University Graduate School of Biomedical Sciences, 3-18-15 Kuramoto-cho, Tokushima 770-8503, Japan; hirao@tokushima-u.ac.jp (A.H.); tanaka.hironori@tokushima-u.ac.jp (H.T.); tomonari.tetsu@tokushima-u.ac.jp (T.T.); hirata.m@tokushima-u.ac.jp (M.H.); bando.masahiro@tokushima-u.ac.jp (M.B.); kida.yoshifumi@tokushima-u.ac.jp (Y.K.); tanaka.takahiro@tokushima-u.ac.jp (T.T.); kawaguchi.tomoyuki@tokushima-u.ac.jp (T.K.); wada.hironori@tokushima-u.ac.jp (H.W.); t-taniguchi@tokushima-u.ac.jp (T.T.); okamoto.koichi@tokushima-u.ac.jp (K.O.); miyamoto.hiroshi@tokushima-u.ac.jp (H.M.); muguruma.naoki@tokushima-u.ac.jp (N.M.); tanahashi@kuh.biglobe.ne.jp (T.T.); 2Department of Community Medicine for Gastroenterology and Oncology, Tokushima University Graduate School of Biomedical Sciences, 3-18-15 Kuramoto-cho, Tokushima 770-8503, Japan; 3Department of Pathophysiology, Tokushima University Graduate School of Biomedical Sciences, 3-18-15 Kuramoto-cho, Tokushima 770-8503, Japan; knishida@tokushima-u.ac.jp

**Keywords:** miR-125b-5p, sorafenib, hepatocellular carcinoma, ataxin-1, drug resistance

## Abstract

**Simple Summary:**

The mechanism of resistance to multikinase inhibitors in hepatocellular carcinoma (HCC) remains unclear. We analyzed miRNA expression profiles in sorafenib-resistant HCC cell lines (PLC/PRF5-R1/R2) and parental cell lines (PLC/PRF5) to identify the responsible miRNAs and target genes involved in the mechanism of resistance. Four miRNAs were significantly upregulated. Among them, we found that miR-125-5p induced sorafenib resistance in HCC cells and in a mouse model. We also revealed that miR-125-5p suppressed ataxin-1 as a target gene and consequently induced Snail-mediated epithelial-mesenchymal transition (EMT) and cancer stemness. Moreover, we demonstrated that ataxin-1 expression has an impact on the prognosis of patients with HCCs. In the future, by comparing the expression status of miR-125b-5p/ataxin-1 and the effect of sorafenib in the clinical setting, it is expected that miR-125b-5p will be established as an effective drug selection marker for treatment selection in patients with HCC.

**Abstract:**

The mechanism of resistance to sorafenib in hepatocellular carcinoma (HCC) remains unclear. We analyzed miRNA expression profiles in sorafenib-resistant HCC cell lines (PLC/PRF5-R1/R2) and parental cell lines (PLC/PRF5) to identify the miRNAs responsible for resistance. Drug sensitivity, migration/invasion capabilities, and epithelial-mesenchymal transition (EMT) properties were analyzed by biochemical methods. The clinical relevance of the target genes to survival in HCC patients were assessed using a public database. Four miRNAs were significantly upregulated in PLC/PRF5-R1/-R2 compared with PLC/PRF5. Among them, miR-125b-5p mimic-transfected PLC/PRF5 cells (PLC/PRF5-miR125b) and showed a significantly higher IC50 for sorafenib compared with controls, while the other miRNA mimics did not. PLC/PRF5-miR125b showed lower E-cadherin and higher Snail and vimentin expression—findings similar to those for PLC/PRF5-R2—which suggests the induction of EMT in those cells. PLC/PRF5-miR125b exhibited significantly higher migration and invasion capabilities and induced sorafenib resistance in an in vivo mouse model. Bioinformatic analysis revealed ataxin-1 as a target gene of miR-125b-5p. PLC/PRF5 cells transfected with ataxin-1 siRNA showed a significantly higher IC50, higher migration/invasion capability, higher cancer stem cell population, and an EMT phenotype. Median overall survival in the low-ataxin-1 patient group was significantly shorter than in the high-ataxin-1 group. In conclusion, miR-125b-5p suppressed ataxin-1 and consequently induced Snail-mediated EMT and stemness, leading to a poor prognosis in HCC patients.

## 1. Introduction

Hepatocellular carcinoma (HCC) is reportedly the fifth most commonly diagnosed malignancy and the second leading cause of cancer-related death worldwide [1]. For patients with unresectable advanced HCC, sorafenib has been the first recommended systemic therapy to demonstrate a survival benefit with an adequate safety profile [2,3]. Sorafenib is an oral tyrosine-kinase inhibitor (TKI) that targets RAF kinase, c-KIT kinase, vascular endothelial growth factor (VEGF) receptors, and platelet-derived growth factor (PDGF) receptors. The phase III SHARP trial showed a median overall survival (OS) of 10.7 months and a disease control rate (DCR) of 43% in the sorafenib treatment group of unresectable HCC patients with well-preserved liver function. The recent phase III REFLECT study demonstrated that lenvatinib was noninferior to sorafenib as a first-line systemic therapy for unresectable HCC [4]. Based on these results, several guidelines positioned sorafenib and lenvatinib as first-line treatments for unresectable HCC. However, currently there is no consensus as to which drug should be used first [5].

To improve OS in unresectable HCC, it is critical to select the appropriate therapeutic agent. Therefore, it is important to identify an appropriate biomarker to distinguish patients who are sensitive to sorafenib. Elucidation of the mechanisms underlying sorafenib resistance and evaluation of resistance factors before treatment would help in selecting an effective drug and developing individualized therapeutic strategies.

In our previous study, we established two sorafenib-resistant cell lines (PLC/PRF5-R1, PLC/PRF5-R2) from PLC/PRF5 cells and reported that high expression of ABCC3 transporter (MRP3) is one of the mechanisms of sorafenib resistance [6]. Moreover, recent studies have revealed that there are several mechanisms underlying acquired resistance to sorafenib, such as activation of PI3K/Akt [7] and JAK–STAT pathways, which are alternative pathways to the MAP kinase signaling pathway [8], activation of hypoxia-inducible pathways [9], and induction of epithelial-mesenchymal transition (EMT) [10]. Although various mechanisms have been suggested, there are no predictive factors that are useful in clinical practice. All of these previous studies utilized liver biopsy tissue to evaluate candidate biomarkers—a method which is difficult to apply clinically. Thus, the development of blood biomarkers is needed for personalized medicine in HCC. 

MicroRNAs (miRNA) are small non-coding RNA molecules of 20–25 nt that regulate gene expression through transcriptional repression and mRNA degradation [11,12,13,14,15]. It has been reported that miRNAs are involved in carcinogenesis, invasion/metastasis, and EMT in various types of cancers [16,17]. For example, miR-125b-5p is abnormally expressed in multiple cancers and is identified as both a tumor promoter and a tumor suppressor in different kinds of cancers [18]. In HCC, miR-125b-5p has been reported to act as a tumor suppressor, exerting inhibitory effects on EMT by small mothers against decapentaplegic (SMAD)2 and 4 [19]. In colorectal cancer, it has been reported that several small non-coding RNAs or miRNAs regulate the epithelial phenotype and EMT by inhibiting the expression of EMT regulators [20]. miRNAs have been used as targets for liquid biopsies, which allow for easy and safe sample collection. Moreover, the development of miRNA microarray assays has made it possible to evaluate the expression of >3000 miRNAs simultaneously [21]. Using this method, Lin and associates reported that miR-378a expression was downregulated in sorafenib-resistant cell lines, and that its target gene IGF1-R was overexpressed, and consequently resistance to sorafenib-induced apoptosis was acquired [22]. However, the association of miR-378a with EMT, cancer stem cell phenotype, and the mechanism of apoptosis resistance to sorafenib is unclear. Moreover, because miR-378a was shown to be downregulated in a resistant cell line, it cannot be used as a biomarker for liquid biopsy. 

EMT is associated with poor patient survival because it is a key step in the development of metastasis in cancer. It has been reported that epithelial cells with EMT lose cell adhesion molecules (such as E-cadherin) and gain mesenchymal cell markers (such as vimentin and Snail), resulting in the loss of polarity and cell-to-cell contacts, enhancement of tumor cell migration and invasion, and resistance to anti-cancer drugs including sorafenib [23,24,25]. Different studies have demonstrated that EMT may be one of the mechanisms of sorafenib resistance [24,26], but the mechanism of EMT regulation remains largely unknown. Therefore, in the process of acquiring resistance to sorafenib, it has been postulated that abnormal miRNA expression is responsible for inducing EMT of HCC, resulting in the acquisition of resistance to sorafenib and enhancement of cell proliferation, metastasis, and invasion capabilities. However, the mechanism by which miRNA regulates EMT in HCC remains poorly understood. 

Ataxin-1 (ATXN1) is a ubiquitous polyglutamine protein expressed primarily in the nucleus where it binds chromatin and interacts with a number of known transcriptional repressors, indicating a role in the regulation of gene expression [27]. ATXN1 loss-of-function is implicated in cancer pathogenesis. In colorectal cancer, it has been reported that ATXN1 is a putative cancer gene and expression of ATXN1 in tumor cells is downregulated compared with normal colon cells [28,29]. However, the full spectrum of ATXN1 functions is far from being fully characterized.

In this study, we investigated miRNA expression profiles in sorafenib-resistant HCC cell lines (PLC/PRF5-R1/R2) to clarify relevant miRNA expression related to sorafenib resistance in comparison with parental PLC/PRF5 cells. We ultimately found upregulation of miRNA125b-5p in resistant cells and identified *ATXN1* as the target gene. We also analyzed the potential role of miRNA125b-5p and ATXN1 in EMT, invasion, migration, and stemness in HCC. Moreover, we used a public database to evaluate the clinical relevance of ATXN1 expression in patients with HCC in association with their survival.

## 2. Materials and Methods

### 2.1. Cell Culture and Compounds

The representative HCC cell line, PLC/PRF5, and Hep3B cells were purchased from American Tissue Culture Collection (ATCC, Manassas, VA, USA). HLF was obtained from the Japanese Cancer Research Resource Bank (JCRB, Osaka, Japan). Cells were grown in Dulbecco’s modified Eagle’s medium (DMEM, Sigma-Aldrich, St. Louis, MO, USA) supplemented with 10% fetal bovine serum (FBS). Sorafenib (LKT Laboratory, St Paul, MN, USA) was dissolved in 100% dimethyl sulfoxide at 40 mM and stored at −20 °C. Sorafenib-resistant cell lines (PLC/PRF5-R1, PLC/PRF5-R2) were established as in our previous report [6]. PLC/PRF5-R1/R2 cells were routinely maintained under constant culture conditions including 10 μM sorafenib. JHH6, HCC cell lines derived from an HCV-positive patient by Dr. Seishi Nagamori, was purchased from JCRB and cultured in Williams E medium (Life Technologies, Gaithersburg, MD, USA). The medium was supplemented with 10% FBS, 2 mM L-glutamine, and 1% penicillin–streptomycin (Life Technologies).

### 2.2. Cell Viability Analysis

Drug sensitivity of cells to sorafenib was estimated using the WST assay as previously described [30]. In brief, cells were seeded in 96-well plates (3 × 10^3^ cells/well) and incubated for 24 h at 37 °C. Sorafenib was then added to the wells at various concentrations and the plates were incubated for another 72 h at 37 °C. Cell viability was measured using a cell counting kit-8 (CCK-8) assay (Dojindo Laboratories, Kumamoto, Japan). CKK-8 solution was added to the wells and the plates were incubated for 3 h at 37 °C. Absorbance at 450 nm was determined using a Spectra Max i3x Platform (Molecular Devices, Inc., Danaher Corporation, Sunnyvale, CA, USA). IC50 values were determined by non-linear regression analysis. Values are presented as the mean ± standard deviation of five experiments.

### 2.3. MicroRNA Microarray Analysis

A human miRNA microarray (based on miRbase release 21.0; Agilent Technologies, Palo Alto, CA, USA) was used for measuring global miRNA expression in cell lines, as previously described [30]. In brief, total RNA was labeled with cyanine 3-cytidine bisphosphate by T4 RNA ligase and hybridized to SurePrint G3 human miRNA microarray release 21.0 using miRNA complete labeling reagent and a hybridization kit (Agilent Technologies). Subsequently, each sample was scanned by a DNA microarray scanner (G2505C; Agilent Technologies), and the fluorescence signal was extracted using feature extraction software (version 10.7.3.1). Raw intensity miRNA data were analyzed using GeneSpring GX version 12 software (Agilent Technologies, Santa Clara, CA, USA).

### 2.4. RT-PCR Analysis

Quantitative real-time PCR was performed as described previously [30]. Total RNA of cell lines, including miRNAs, was extracted using a Qiagen miRNeasy Mini kit (Qiagen, Hilden, Germany). RNA was then reverse transcribed into cDNA using a High Capacity RNA-to-cDNA kit (Applied Biosystems, Waltham, MA, USA). Quantitative real-time PCR was then carried out in 96-well plates in a StepOnePlus Real-Time PCR System (Applied Biosystems) using TaqMan Universal PCR Master Mix (Applied Biosystems) to monitor the PCR amplification. The real-time PCR mixtures using TaqMan Universal PCR Master Mix consisted of 5.0 μL of TaqMan Gene Expression Master Mix (2×) and 0.5 μL TaqMan Gene Expression Assay (20×) (Appendix A). The following two-step cycling program was used for PCR reactions: 50 °C for 2 min and 95 °C for 10 min, and 40 cycles of (95 °C, 15 s; and 60 °C, 1 min). All samples were amplified in triplicate and relative quantification of the expression level of each gene was calculated. *GAPDH* was used as the endogenous reference gene.

### 2.5. Overexpression and Knockdown Experiments

The miRNA mimics and random miRNA used as a control were purchased from Thermo Fisher Scientific (Waltham, MA, USA) (Appendix A). Each miRNA was transfected into cells at a concentration of 10 nM, as we previously described [31]. Sorafenib was added to the cells after 24 h, followed by incubation for 72 h. Cell viability was then determined by WST assay. *ATXN1* expression plasmid or negative control vector (vector NC) (Appendix A) was transiently transfected to the cells using Lipofectamine 3000 (Invitrogen, Carlsbad, CA, USA) according to the manufacturer’s instructions. For knockdown experiments, cells were transfected with 10 nM of *ATXN1* specific small interfering RNA (siRNA) or random siRNA as a control (Appendix A). Sorafenib was added to the cells after 24 h, followed by incubation for 72 h. Cell viability was then determined by WST assay. 

### 2.6. In Silico Identification of miRNA Target Genes

Target genes were bioinformatically predicted based on the miRNA seed sequence by using miRDB [32], an online database for miRNA target prediction and functional annotations [33], and Target Scan [34].

### 2.7. miRNA Luciferase Reporter Assay

Double-stranded oligonucleotides for the 3′-UTR of *ATXN1* harboring miR-125b-5p binding sites 1 and 2, respectively, were prepared by heating equal amounts of complementary single strands at 95 °C for 15 min and gradually cooling to room temperature (Appendix A). The DNA fragments were subcloned into psiCHECK-2 vector (Promega, Madison, WA, USA) using XhoI and NotI restriction sites. The plasmids with two mutant types for each miR-125b-5p binding site were also prepared by replacing six base pairs at the 3′-UTR of the seed sequence.

HEK293T (ATTC, #CRL-3216) cells (8.0 × 10^4^) were cultured on 24-well plates. After 24 h incubation, miR-125b-5p (10 nM) or control mimic miRNA (10 nM) was transfected. Twelve hours after the transfection of mimic miRNAs, psiCHECK-2 constructs with various site-directed mutations were transfected. Another 12 h after the second transfection, cells were harvested and the firefly and Renilla luciferase activities were measured using the Dual-Luciferase Reporter Assay System (Promega, Madison, WI, USA).

### 2.8. Promoter Assay

The 5′-flank of the human *ATXN1* gene was cloned into the pGL4.21-basic luciferase reporter vector (Promega, Madison, WI, USA). In brief, the first PCR was performed using human genomic DNA as a template. The *ATXN1* proximal promoter region between 834 bp upstream and 151 bp downstream from the transcriptional start site was amplified using primer sets listed in Appendix A. The amplified products were subcloned into the pGL4.21-basic vector using NheI and HindIII restriction sites. HEK293T cells (8.0 × 10^4^) were cultured on 24-well plates. After 24 h incubation, miR-125b-5p (10 nM) or control mimic miRNA (10 nM) was transfected. Twelve hours after the transfection of mimic miRNAs, pGL-4.21 luciferase constructs with or without the promoter region (100 ng) were cotransfected with pGL4.74 vector (50 ng) using X-tremeGENE HP DNA transfection reagent (Roche, Basel, Switzerland). Twenty-four hours after the transfection, cells were harvested, and the firefly and Renilla luciferase activities were measured using the Dual-Luciferase Reporter Assay System (Promega, Madison, WI, USA).

### 2.9. Western Blot Analysis

Expression of proteins was analyzed by western blot analysis, as described previously [35]. Briefly, cells were washed with phosphate buffered saline (PBS) and lysed in RIPA buffer containing protease inhibitors (Sigma-Aldrich, Tokyo, Japan). Cell lysates were analyzed for protein content, resolved by sodium dodecyl sulfate polyacrylamide gel electrophoresis (SDS–PAGE), and transferred to polyvinylidene fluoride membranes using a semi-dry transfer apparatus (Bio-Rad, Hercules, CA, USA). Blots were blocked with 5% fat-free dry milk in Tris-buffered saline with Tween 20 (TBS–T) for 1 h and then incubated overnight with rabbit anti-human E-cadherin monoclonal antibody, rabbit anti-vimentin monoclonal antibody, rabbit anti-human Snail monoclonal antibody (Cell Signaling Technology, Tokyo, Japan), mouse anti-ATXN1 monoclonal antibody (Santa Cruz Biotechnology, Dallas, TX, USA), and mouse anti-β-actin monoclonal antibody (Sigma-Aldrich, Tokyo, Japan) as primary antibodies. The membranes were washed with TBS–T and incubated with secondary horseradish conjugated sheep anti-mouse antibody or donkey anti-rabbit antibody (GE Healthcare UK Limited, Buckinghamshire, UK). The proteins were visualized by standard procedures including an ECL detection system (GE Healthcare UK Limited). To ensure equal protein loading, the same blot was developed for β-actin (Sigma-Aldrich, Tokyo, Japan) as a loading control. The expression levels of each protein were quantified by densitometric analysis using Image Lab Software (Bio-Rad, Hercules, CA, USA).

### 2.10. Wound Healing Assay

Cells (1.0 × 10^6^) for the wound healing assay were seeded in 6-well plates. When the cell confluence reached about 80%, scratch wounds were made by scraping the cell layer across each culture plate using the tip of a 200 µL pipette. After wounding, the debris was removed by washing the cells with PBS. Wounded cultures were incubated in the culture medium containing 10% FBS with miRNA mimic or siRNA. Wound healing was imaged every 24 h for 72 h using a BZ-X710 fluorescence microscope (Keyence, Osaka, Japan). Wound healing ability was determined by measuring the mean migration distance, which was calculated by dividing the wound repair area by the width. Experiments were carried out three times. 

### 2.11. Cell Invasion Assay

Cell invasion assays were performed as described previously [31]. In brief, cells transfected with miRNA mimic or siRNA were seeded at 5 × 10^5^ cells/well in the upper chamber of a CytoSelect 24-Well Cell Invasion Assay kit (Cell Biolabs, San Diego, CA, USA). The number of invading cells was counted in three randomly selected views under a microscope. 

### 2.12. Flow Cytometry Analysis

CD44 and CD133 are well-known cancer stem cell (CSC) surface markers commonly used in HCC [36]. Cells were tested for CD44 and CD133 by incubating them with mouse anti-human CD44 monoclonal antibody (APC) (17044181, Thermo Fisher Scientific) and mouse anti-human CD133 monoclonal antibody (PE) (372803, BioLegend, San Diego, CA, USA). After washing the cells two times with PBS containing 1% BSA and 1% sodium azide, the expression of CD44 and CD133 was assessed using a BD FACS Verse flow cytometer (BD Biosciences, San Jose, CA, USA). Annexin V and propidium iodide (PI) were used for the detection of apoptosis.

### 2.13. Tumor Xenograft Experiments

Female 6-week-old BALB/c nu/nu mice (CLEA Japan Inc., Tokyo, Japan) were used for all studies. PLC/PRF5 cells were infected with MISSION^®^ Lenti microRNA expressing miR125b-5p or negative control virus (Appendix A) according to the manufacturer’s instructions. Tumor cells (0.5 × 10^7^) were then inoculated into the flanks of mice. Sorafenib or vehicle control was administered orally, once a day, for 21 days at dose levels of 30 mg/kg, as reported previously [37]. Treatment began when tumors reached a volume of 200 mm^3^. Tumor size was calculated using the equation (l × w^2^)/2, where l and w refer to the larger and smaller dimensions collected at each measurement. Tumor dimensions were recorded three times a week starting on the first day of treatment. All animal experiments were carried out according to the Guidelines for Animal Experiments at Tokushima University.

### 2.14. ATXN1 Expression from RNA-Chip Data

The profiling expression of *ATXN1* in HCC tissues from microarray studies was acquired from the Genomic Data Commons data portal [38]. Three studies (TCGA–LIHC [39], GSE76427 [40], and GSE10141 [41]) were available for analyzing both *ATXN1* mRNA expression and useable survival data. However, since GSE76427 and GSE10141 contained small study populations (24 and 80 patients, respectively), only data from TCGA–LIHC (377 patients) were used. Fragments Per Kilobase Million with Upper Quantile (FPKM–UQ) values normalized by RNA-seq counts were used to perform survival analysis. The cutoff value was set at 50,000 FPKM–UQ based on the median *ATXN1* mRNA expression; cases with ≥50,000 FPKM–UQ were designated as the high expression group, and cases with lower FPKM–UQ (<50,000) as the low expression group. OS in the high- versus low-*ATXN1* expression group was compared using the Kaplan–Meier method and log-rank test.

### 2.15. Statistical Analysis

Data are presented as mean ± SD. The statistical significance of the difference between the values of the control and treatment groups was evaluated by either Student’s *t*-test or ANOVA, followed by Dunnett’s test using Prism version 5 (GraphPad Software, San Diego, CA, USA). Values of *p* < 0.05 were considered statistically significant.

## 3. Results

### 3.1. Differential miRNA Expression Profile between Sorafenib-Resistant PLC/PRF5-R1/R2 and PLC/PRF5 Cells

In our previous study, we established two sorafenib-resistant cell lines (PLC/PRF5-R1, PLC/PRF5-R2) from PLC/PRF5 cells [6]. To identify miRNAs involved in sorafenib resistance for HCC cells, we first performed miRNA microarray analysis and compared miRNA expression profiles of PLC/PRF5-R1 and PLC/PRF5-R2 cells with those of parental PLC/PRF5 cells. When analyzing the upregulated miRNAs in sorafenib-resistant cells (fold change > 2, signal intensity > 100), PLC/PRF5-R1 and PLC/PRF5-R2 showed higher expression of six miRNAs and seven miRNAs, respectively, compared with parental PLC/PRF5. (Figure 1a; Appendix A). The common four miRNAs showing increased expression in the two resistant cell lines were miR-100-5p, miR-125b-5p, miR-193b-3p, and miR-210-3p (Figure 1a). To validate the expressions of those miRNAs, we performed RT-PCR and found significantly higher expression levels of miR-100-5p, miR-125b-5p, miR-193b-3p, and miR-210-3p in PLC/PRF5-R1 and PLC/PRF5-R2 cells compared with parental PLC/PRF5 cells (Appendix A).

In the analysis of downregulated miRNAs, PLC/PRF5-R1 and PLC/PRF5-R2 showed lower expressions of three miRNAs (miR-192-5p, miR-194-5p, and miR-215-5p) compared with PLC/PRF5 cells (Figure 1b). RT-PCR also revealed significantly lower expression levels of miR-192-5p, miR-194-5p, and miR-215-5p in PLC/PRF5-R1 and PLC/PRF5-R2 cells compared with parental PLC/PRF5 cells (Appendix A).

### 3.2. miR-125b-5p Confers Resistance to Sorafenib in PLC/PRF5 Cells

To investigate the relevance of upregulated miRNAs in sorafenib resistance, we transfected mimics of miR-100-5p, miR-125b-5p, miR-193b-3p, and miR-210-3p into parental PLC/PRF5 cells and examined the change in sensitivity to sorafenib in those transfected cells by calculating the IC50 by WST assay. The expression levels of these four miRNAs were 3 × 10^3^ to 4 × 10^5^-fold higher in PLC/PRF5 cells transfected with their mimics compared with those in control cells by RT-PCR, indicating high transfection efficiency (Figure 2a, Appendix A). Of the four miRNAs, miR-125b-5p mimic-transfected PLC/PRF5 cells (PLC/PRF5-miR125b) showed significantly higher IC50 values for sorafenib (6.59 μM, 95%Cl: 6.40–6.81 μM) compared with those transfected with negative control miRNA mimic (PLC/PRF5-miNC) (5.05 μM (95%Cl: 4.83–5.29 μM); *p* < 0.05) (Figure 2b), whereas transfectants of other mimics did not show any significant change in IC50 values compared with PLC/PRF5-miNC (Appendix A). 

In the experiments examining downregulated miRNAs, we transfected miR-192-5p, miR-194-5p, or miR-215-5p mimics in PLC/PRF5-R2 cells to determine whether these miRNAs rescue the resistance to sorafenib. The expression levels of these three miRNAs were 2 × 10^4^ to 4 × 10^4^-fold higher in PLC/PRF5-R2 cells transfected with their mimics compared with control cells, indicating high transfection efficiency (Appendix A). However, PLC/PRF-R2 cells transfected with those mimics did not show any significant change in IC50 values compared with PLC/PRF5-R2 cells transfected with negative control miRNA (PLC/PRF5-R2-miNC), indicating that these miRNAs do not have any effects on impairment of resistance to sorafenib (Appendix A). Accordingly, we selected miR-125b-5p for further experimental analysis. 

### 3.3. miR-125b-5p Induces EMT in PLC/PRF5 Cells

Since drug resistance is one of the major phenotypes associated with EMT, we investigated expression of EMT-related proteins in PLC/PRF5-miR125b cells and sorafenib-resistant cells (PLC/PRF5-R2-miNC) in comparison with PLC/PRF5-miNC by western blot analysis. The expression of E-cadherin in PLC/PRF5-R2-miNC cells was obviously lower compared with PLC/PRF5-miNC cells, whereas expression of Snail and vimentin in PLC/PRF5-R2-miNC cells was obviously higher compared with PLC/PRF5-miNC cells. In PLC/PRF5-miR125b cells, the expression of E-cadherin was slightly lower and expression of Snail and vimentin was higher compared with PLC/PRF5-miNC cells (Figure 2c). Thus, the expression of EMT-related proteins in PLC/PRF5-miR125b cells was changed toward that of PLC/PRF5-R2 cells. 

In the microscopic analysis of the morphology, PLC/PRF5-R2 cells showed a fibroblast-like shape, which was morphologically distinct from the round or oval shape of PLC/PRF5-miNC cells—findings characteristic of epithelial cells—suggesting that PLC/PRF5-R2 cells acquired a mesenchymal phenotype. The morphology of PLC/PRF5-miR125b cells was distinct from that of PLC/PRF5-miNC cells, but similar to PLC/PRF5-R2 cells (Figure 2d). Taken together, these results indicate that miR-125b-5p has EMT-inducing ability in PLC/PRL5 cells, leading to acquisition of sorafenib resistance.

### 3.4. Predicted Target Genes of miR-125b-5p

To elucidate the underlying regulatory mechanism of miR-125b-5p on EMT, we applied miRDB to screen potential targets of miR-125b-5p and selected the top 50 genes sorted by the target scores (Appendix A). Among them, we chose the genes that negatively controlled EMT and were also downregulated in PLC/PRF5-R1/R2 cells compared with PLC/PRF5 cells. As a result, we identified the *ATXN1* gene as a potential target of miR-125b-5p. Target Scan predicted two nucleotide sequences (position 2439-2445, 4097-4104) in the 3′-UTR of the *ATXN1* gene matching the miR-125b-5p sequence (Figure 3a). To verify the potential binding sequences of miR-125b-5p, we prepared the reporter plasmids (psiCHECK2) that expressed chimeric RNAs containing the sequence for Renilla and 3′-UTR sequence of *ATXN1* and evaluated the effects of miRNA on the chimeric RNAs using the dual luciferase assay system. The overexpression of miR-125b-5p significantly reduced the luciferase activity of Renilla_*ATXN1*-3′-UTR (Figure 3b). Two mutant vectors (MT1 and MT2), which harbored seven-point mutations within each binding site for miR-125b-5p (Figure 3a), did not show a change in luciferase activity (Figure 3b). To reveal whether miR-125b-5p regulates the transcription of *ATXN1*, we assessed the promoter activity of *ATXN1* using a dual-luciferase reporter assay. Overexpression of miR-125b-5p resulted in no decrease in luciferase activity (Figure 3c). These results suggested that miR-125b-5p does not regulate the transcription of *ATXN1*, but is involved in translational regulation including RNA degradation.

The *ATXN1* mRNA level in PLC/PRF5-miR125b determined by RT-PCR was decreased to 34% of control cells (*p* < 0.01; Figure 3d). The protein expression of ATXN1 in PLC/PRF5-miR125b cells was lower than in PLC/PRF5-miNC (Figure 3e).

Since ATXN1 is known to suppress the transcription of Snail [42], we transfected *ATXN1* siRNA (siATXN1) or negative control siRNA (siNC) in PLC/PRF5 cells (PLC/PRF5-siATXN1 and PLC/PRF5-siNC) and examined changes in Snail expression. The *ATXN1* mRNA level in PLC/PRF5-siATXN1 cells was suppressed to 16% of the level in PLC-PRF5-siNC cells (Appendix A). The mRNA level of *Snail* in PLC/PRF5-siATXN1 cells was significantly higher compared with PLC/PRF5-siNC cells (*p* < 0.01; Figure 3f). These results indicate that *ATXN1* is the target gene of miR-125b-5p and that knocking down *ATXN1* directly upregulates *Snail*, possibly leading to promotion of EMT and acquisition of drug resistance (Figure 3g).

### 3.5. Downregulation of ATXN1 Induces EMT and Sorafenib Resistance

To examine the role of ATXN1 in EMT and drug resistance, we next examined EMT-related protein expression, morphology, and sensitivity to sorafenib in PLC/PRF5-siATXN1 cells compared with those in PLC/PRF5-siNC and PLC/PRF5-R2 cells transfected with siRNA negative control (PLC/PRF5-R2-siNC). The protein expression of ATXN1 in PLC/PRF5-siATXN1 and PLC/PRF5-R2-siNC was substantially lower than that of control cells. The expression of Snail and vimentin in PLC/PRF5-siATXN1 and PLC/PRF5-R2-siNC cells was obviously higher compared with PLC/PRF5-siNC cells, whereas the expression of E-cadherin in those cells was lower compared with PLC/PRF5-siNC cells (Figure 4a). The morphology of PLC/PRF5-siATXN1 cells was distinct from that of PLC/PRF5-siNC cells; the latter were adhesively contacted with each other and showed an oval or round shape, but the former lost cell-cell adhesion and had a non-round shape—findings similar to those for PLC/PRF5-miR125b cells (Figure 4b). These results suggest that knockdown of the *ATXN1* gene induced EMT in PLC/PRF5 cells.

The IC50 value of sorafenib against PLC/PRF5-siATXN1 was significantly higher compared with PLC/PRF5-siNC cells (6.87 μM (95%Cl: 6.49–7.47 μM) vs. 5.26 μM (95%Cl: 4.89–5.46 μM); *p* < 0.05) (Figure 4c). Thus, miR-125b-5p negatively regulated ATXN1, causing Snail-mediated EMT, consequently leading to drug resistance. 

In the flow cytometry analysis, the percentage of annexin V (+) in PLC/PRF5-miR125b cells (41.3 ± 1.3%) was significantly lower compared with control cells treated with mimic negative control (46.8 ± 0.6%; *p* < 0.05) (Appendix A). Similarly, the percentage of annexin V (+) in PLC/PRF5-siATXN1 cells (53.4 ± 2.3%) was significantly lower than that with siRNA negative control (59.4 ± 1.4%, *p* < 0.05) (Appendix A).

### 3.6. miR-125b-5p and ATXN1 Modulate Cell Migration and Invasion

Because EMT is reportedly associated with cancer migration and invasion [43], we examined the effects of miR-125b-5p mimic and siATXN1 on migration and invasion of PLC/PRF5 cells using a wound healing assay and invasion assay, respectively. Representative images from the wound healing assay in PLC/PRF5-miNC, PLC/PRF5-miR125b, and PLC/PRF5-R2-miNC cells are shown (Figure 5a). Quantitative analysis of images indicated that the migration distance in PLC/PRF5-miR125b (599 ± 55 μm) and PLC/PRF5-R2-miNC cells (675 ± 72 μm) was significantly higher than in PLC/PRF5-miNC cells (381 ± 46 μm; *p* < 0.05) (Figure 5a). Similarly, representative images from the wound healing assay in PLC/PRF5-siNC, PLC/PRF5-siATXN1, and PLC-PRF5-R2-siNC cells are shown (Figure 5b). Quantitative analysis of the images indicated that the migration distance in PLC/PRF5-siATXN1 (735 ± 25 μm) and PLC/PRF5-R2-siNC cells (781 ± 44 μm) was significantly higher than in PLC/PRF5-siNC cells (426 ± 36 μm; *p* < 0.05) (Figure 5b). The results suggest that miR-125b-5p could induce CSCs through inhibition of *ATXN1*.

Representative images from the invasion assay are shown (Figure 5c,d). Quantitative analysis revealed that the number of invading cells in PLC/PRF5-miR125b (23.7 ± 2.5) and PLC/PRF5-R2-miNC (24.3 ± 4.0) was significantly higher than in PLC/PRF5-miNC cells (10 ± 1.4; *p* < 0.05) (Figure 5c). Similarly, the number of invading cells in PLC/PRF5-siATXN1 (26 ± 4.2) and PLC-PRF5-R2-siNC (16 ± 3.7) was significantly higher than in PLC/PRF5-miNC cells (5.7 ± 2.5; *p* < 0.05) (Figure 5d). These data suggest that miR-125b-5p enhanced cell migration and invasion capabilities through inhibition of *ATXN1*.

### 3.7. miR-125b-5p and Downregulation of ATXN1 Confer Stemness Characteristic in HCC Cells

Since EMT is reportedly associated with CSC generation, which plays a critical role in drug resistance [44], we performed a flow cytometric analysis of CSC markers in PLC/PRF5-miR125b and PLC/PRF5-siATXN1 in comparison with control cells. In the flow cytometry analysis of CD44 and CD133, the double positive cell subpopulation in PLC/PRF5-miR-125b (4.22 ± 1.1%) was significantly higher than in PLC/PRF5-miNC cells (1.67 ± 0.44%; *p* < 0.05), suggesting that miR-125b-5p enhances CSC population (Figure 5e). Similarly, the double positive cell subpopulation in PLC/PRF5-siATXN1 (4.41 ± 1.37%) was significantly higher than in PLC/PRF5-siNC cells (1.48 ± 0.49%; *p* < 0.05) (Figure 5f). 

### 3.8. miR-125b-5p and siATXN1 Confer Drug Resistance by Inducing EMT in Various HCC Cell Lines

To verify the function of miR-125b-5p in the other HCC cell lines, we transfected miR-125b-5p or siATXN1 into Hep3B cells (hepatitis B virus-related HCC cell line; Hep3B-miR125b, Hep3B-siATXN1) and examined drug sensitivity and expression of EMT-related proteins. The IC50 of sorafenib against Hep3B-miR125b was significantly higher compared with Hep3B cells transfected with miNC (Hep3B-miNC) (2.71 μM (95%Cl: 2.34–3.17 μM) vs. 1.28 μM (95%Cl: 0.95–1.65 μM); *p* < 0.05) (Figure 6a). Moreover, the IC50 of sorafenib against Hep3B-siATXN1 was significantly higher compared with Hep3B transfected with siNC (Hep3B-siNC) (2.42 μM (95%Cl: 1.91–3.01 μM) vs. 1.41 μM (95%Cl: 1.19–1.65 μM), *p* < 0.05) (Figure 6b). In the western blot analysis of EMT-related proteins, the expression of ATXN1 in Hep3B-miR125b and Hep3B-siATXN1 cells was substantially lower than in control cells. The expression of Snail and vimentin in those cells was higher, and the expression of E-cadherin was slightly lower than in control cells (Figure 6c,d). These results strongly suggest that miR-125b-5p conferred sorafenib resistance and EMT in Hep3B cells.

Next, we transfected miR-125b-5p or siATXN1 into JHH6 cells (derived from hepatitis C virus positive HCC; JHH6-miR125b, JHH6-siATXN1). The IC50 of sorafenib against JHH6-miR125b was significantly higher compared with JHH6 cells transfected with miNC (JHH6-miNC) (8.36 μM (95%Cl: 8.09–8.67 μM) vs. 7.29 μM (95%Cl: 6.77–7.46 μM); *p* < 0.05) (Figure 6e). Moreover, the IC50 of sorafenib against JHH6-siATXN1 was significantly higher compared with JHH6 transfected with siNC (JHH6-siNC) (9.02 μM (95%Cl: 8.58–9.31 μM) vs. 7.76 μM (95%Cl: 7.12–8.03 μM), *p* < 0.05) (Figure 6f). The expression of ATXN1 in JHH6-miR125b and JHH6-siATXN1 cells was substantially lower than in control cells. The expression of Snail and vimentin in those cells was higher, and the expression of E-cadherin was lower than in control cells (Figure 6g,h). The morphology of JHH6-miR125b and JHH6-siATXN1 cells was distinct from that of JHH6-miNC and JHH6-siNC cells; the latter were adhesively contacted with each other and showed an oval or round shape, but the former lost cell-cell adhesion and had a non-round shape—findings similar to those for PLC/PRF5-miR125b cells (Appendix A). These results strongly suggest that the miR-125b-5p-ATXN1 axis conferred sorafenib resistance and EMT in HCC cell lines with different etiologies.

### 3.9. Overexpression of ATXN1 Confer Drug Resistance and EMT in HCC Cells

To verify the function of ATXN1, PLC/PRF-R2 cells and HLF cells, which possess a more mesenchymal phenotype [45], were transfected with *ATXN1* expressing the plasmid vector or negative control vector (PLC/PRF5-R2-ATXN1, PLC/PRF5-R2-vector NC, HLF-ATXN1, and HLF-vector NC, respectively). The IC50 of sorafenib against PLC/PRF5-R2-ATXN1 cells was significantly lower compared with PLC/PRF5-R2-vector NC (14.9 μM (95%Cl: 9.66–17.03 μM) vs. 16.53 μM (95%Cl: 14.99–17.96 μM); *p* < 0.05) (Figure 7a). Moreover, the IC50 of sorafenib against HLF-ATXN1 cells was significantly lower compared with HLF-vector NC (7.81 μM (95%Cl: 5.99–8.80 μM) vs. 10.74 μM (95%Cl: 9.69–11.89 μM), *p* < 0.05) (Figure 7b). The expression of Snail and vimentin in those cells was lower, and the expression of E-cadherin was higher than in control cells (Figure 7c,d). The morphology of PLC/PRF5-R2-ATXN1 and HLF-ATXN1 cells were more epithelial-like, adhesively contacted with each other, and showed an oval or round shape (Appendix A). These results strongly suggest that ATXN1 abrogated sorafenib resistance and EMT in PLC/PRF5-R2 and HLF cells.

### 3.10. Overexpression of miR-125b-5p Confer Sorafenib Resistance In Vivo

To investigate whether miR-125b-5p could promote sorafenib resistance in vivo, we established a xenograft mouse model. After establishing PLC/PRF5 cells stably expressing miR-125b-5p using lentiviral vectors, cells were inoculated subcutaneously in the dorsal flank of nude mice. After 21 sorafenib treatments (30 mg/kg/day), the volume of the experimental group was significantly larger than that in controls (*n* = 6, 773 ± 145 mm^3^ vs. 427 ± 84 mm^3^, *p* < 0.05). These results suggested that overexpression of miR-125b-5p increased the resistance to sorafenib in vivo (Figure 8a,b).

### 3.11. ATXN1 Expression was Significantly Associated with Survival in Patients with Advanced HCC

To investigate the clinical relevance of ATXN1 expression in patients with advanced HCC, we performed a Kaplan–Meier analysis to determine the association between ATXN1 expression and OS using the TCGA–LIHC dataset. Patients with Stage III and IV HCC (*n* = 91) were chosen and 90 patients with ATXN1 data available were divided into low-ATXN1 and high-ATXN1 groups (Figure 9a). The low-ATXN1 group exhibited significantly shorter survival than the high-ATXN1 group (*p* = 0.0153; Figure 9b). The median OS in the low-ATXN1 group was much shorter than in the high-ATXN1 group (18.35 vs. 39.78 months, hazard ratio 0.49, 95%Cl: 0.28–0.87; *p* < 0.05).

## 4. Discussion

In this study, we used miRNA microarray analysis to identify 125b-5p as a pivotal miRNA underlying sorafenib resistance in HCC cells. Using in vitro and in vivo approaches, we found that miR-125b-5p directly inhibited *ATXN1* gene expression and consequently induced EMT through enhanced Snail expression, leading to resistance to sorafenib. We also found that miR-125b-5p and downregulation of *ATXN1* enhanced migration/invasion capability and stemness of HCC cells, and that overexpression of *ATXN1* reverses EMT and sorafenib resistance. Furthermore, patients with HCC in the low-ATXN1 expression group exhibited a significantly shorter survival time than those in the high-ATXN1 group, as indicated by Kaplan–Meier analysis. This is the first report to show a pivotal role of the miR-125b-5p-ATXN1 axis in the process of EMT, including acquisition of drug resistance, in HCC. 

Our results showing that overexpression of miR-125b-5p was closely associated with EMT and drug resistance in sorafenib-resistant HCC cells are consistent with previous studies in other types of cancers. Lu and associates reported that miR-125b-5p augmented cetuximab resistance through activation of the Wnt signaling pathway in colorectal cancer [46]. Similarly, Yu and associates reported that miR-125b-5p enhanced 5-FU resistance in colorectal cancer [47]. In contrast, Zhang and associates reported that decreased expression of miR-125b-5p promoted EMT and conferred resistance to paclitaxel in non-small cell lung cancer [48]. These conflicting results have been explained by differences in the target genes among different types of cancer cells. In addition, Jun-Nian and associates reported that miR-125b-5p exerts inhibitory effects on EMT and EMT-associated traits in HCC by inhibiting *SMAD2* and *SMAD4* mRNA expression [19]. However, in our analysis of PLC/PRF5-R2 cells, *SMAD2* mRNA expression was conversely upregulated, and *SMAD4* was unchanged compared with PLC/PRF5 cells despite higher expression of miR-125b-5p (Appendix A). Importantly, we were able to show that the miR-125b-5p-ATXN1 axis is associated with sorafenib resistance as well as EMT in various HCC cell lines, including PLC/PRF5 and Hep3B which were infected with HBV and JHH-6, which was derived from an HCV-positive patient. 

We identified the ATXN1 gene as a target gene of miR-125b-5p in this study. The role of ATXN1 in cancers has long remained unclear. However, Kang and associates reported that ATXN1 inhibited Snail expression, which is a transcription factor for E-cadherin, leading to EMT in cervical cancer [43]. In other words, reduced ATXN1 expression caused EMT in cervical cancer cells. Our data indicated that miR-125b-5p inhibited ATXN1 expression and, therefore, induced Snail-mediated EMT, resulting in augmented drug resistance, enhanced invasion/migration capability, and a stem cell phenotype. Furthermore, overexpression of ATXN1 reversed EMT and sorafenib resistance. Thus, it is suggested that ATXN1 expressed in cancer cells protects the cells from EMT and subsequent malignant progression. This is also supported by our clinical data showing that the OS time in the low-ATXN1 HCC patient group was shorter than in the high-ATXN1 HCC patient group. 

Various mechanisms for acquiring resistance to sorafenib in HCC have been reported to date. Activation of PI3K/Akt [7] and JAK–STAT pathways [8] due to crosstalk with RAS/RAF/MEK pathways, activation of hypoxia-inducible pathways [49], and EMT [24] have been implicated. Among these, EMT, which we observed in this study, is closely associated with the malignant transformation of cancer, and it is reportedly associated with acquisition of invasive capacity, drug resistance, and increased cell proliferation [50,51,52]. Thus, EMT can be assumed to be one of the major mechanisms of resistance to sorafenib. Previously, we reported that the ABC transporter MRP3 is highly expressed in the sorafenib-resistant cell line used in this study [6]. It has been reported that epithelial cells, which undergo EMT, overexpress not only mesenchymal biomarkers but also CSC markers [53]. Furthermore, CSCs have been reported to acquire drug resistance through increased expression of ABC transporters [52,54]. In fact, we found that the transfection of miR-125b-5p into PLC/PRF5 cells caused MRP3 gene upregulation (Appendix A). Thus, it was assumed that the increased expression of MRP3 in sorafenib-resistant (PLC/PRF5-R1, PLC/PRF-R2) cells could be mediated by EMT-associated CSCs.

Systemic chemotherapy is recommended for HCC patients with Barcelona Clinic Liver Cancer (BCLC) Stage C. In addition to sorafenib, lenvatinib and a combination of atezolizumab and bevacizumab, are currently recommended as first-line therapy. Regorafenib, cabozantinib, ramucirumab, nivolumab, ipilimumab, and pembrolizumab are recommended as second-line therapy. In contrast, it is recommended that patients with BCLC stage B undergo transcatheter arterial chemoembolization (TACE) as standard care. Although it has been recently reported that TKIs are preferred over TACE to maintain hepatic functional reserve [55], guidelines for choosing among these drugs are lacking. Since the miR-125b-5p-ATXN1 axis has been implicated in sorafenib resistance and EMT, it may be useful as a biomarker for drug selection in systemic therapy and the choice between TACE and TKI therapy as an initial treatment. 

To date, there are no available reports investigating the effect of miR-125b-5p on the efficacy of sorafenib in clinical practice. Therefore, our finding needs to be validated clinically, ideally in prospective trials, by comparing the expression status of miR-125b-5p and the effect of sorafenib in patient samples to see if our finding can be validated in real life and determine that it is not simply a cell-line phenomenon, which could lead to miR-125b-5p being established as an effective drug selection marker for patients with HCC.

## 5. Conclusions

In conclusion, the present study clarified the mechanism of sorafenib resistance in HCC cells, in which miR-125b-5p suppresses ATXN1 and induces Snail-mediated EMT and CSCs. Moreover, we demonstrated that ATXN1 expression has an impact on the prognosis of patients with HCC.

## Figures and Tables

**Figure 1 cancers-13-04917-f001:**
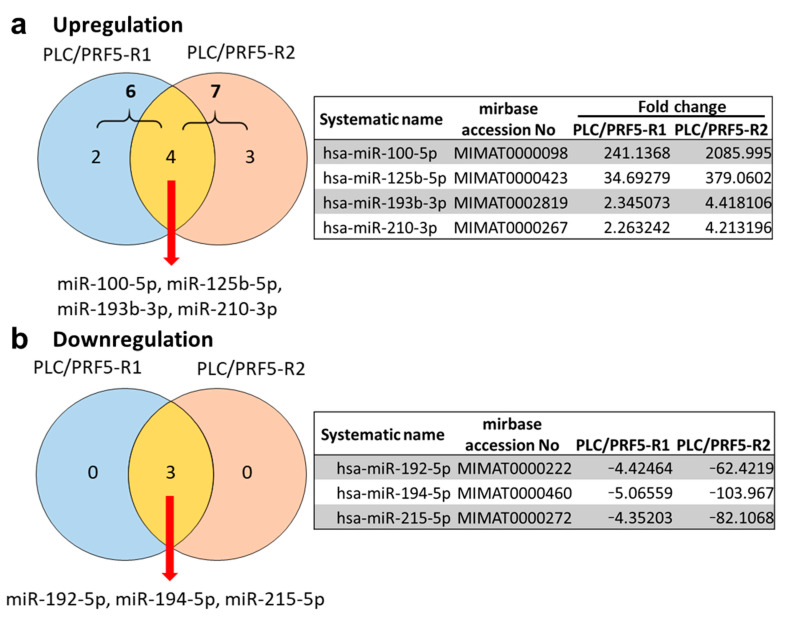
Differentially expressed miRNAs in sorafenib-resistant hepatocellular carcinoma (HCC) cell lines PLC/PRF5-R1 and PLC/PRF5-R2. The miRNA expression profile was compared between PLC/PRF5-R1 or PLC/PRF5-R2 and parental PLC/PRF5. Venn diagram of the differentially expressed miRNAs in PLC/PRF5-R1 and PLC/PRF5-R2 cells (fold change > 2, signal intensity > 100), and the miRNA list identified in the common part of the Venn diagram are shown. (**a**) Upregulated miRNAs. (**b**) Downregulated miRNAs.

**Figure 2 cancers-13-04917-f002:**
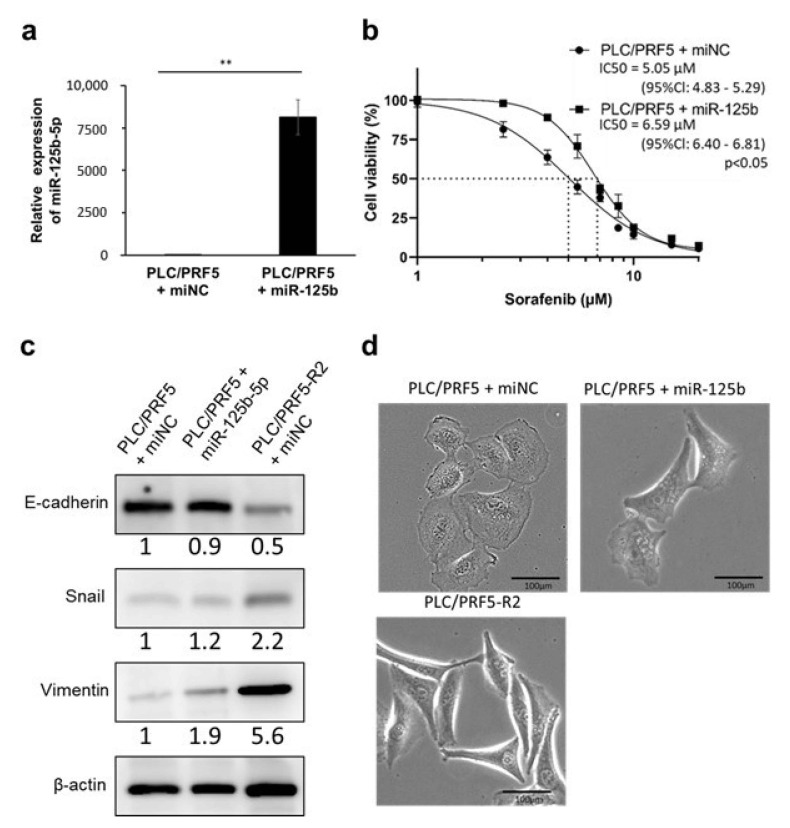
Forced expression of miR-125b-5p confers sorafenib resistance and epithelial-mesenchymal transition (EMT) phenotype in PLC/PRF5 cells. (**a**) The relative expression levels of miR-125b-5p in PLC/PRF5 cells transfected with miR125b-5p mimic (PLC/PRF5-miR125b) and PLC/PRF5 cells transfected with negative control miRNA (PLC/PRF5-miNC) were determined by quantitative real-time PCR (RT-PCR). ** *p* < 0.01. (**b**) The cell viability of PLC/PRF5-miR125b and PLC/PRF5-miNC cells treated with various concentrations of sorafenib was determined by WST assay. (**c**) Expression of EMT markers (E-cadherin, Snail, vimentin) in PLC/PRF5-miNC, PLC/PRF5-miR-125b, and PLC/PRF5-R2 cells was examined by western blot analysis. Densitometry analysis indicates relative protein levels from one representative of three independent experiments. Numbers below the bands represent protein expression normalized to β-actin. (**d**) Representative images of PLC/PRF5-miNC, PLC/PRF5-miR125b, and PLC/PRF5-R2 cells under observation with a stereoscopic microscope are shown. Scale bars, 100 μm.

**Figure 3 cancers-13-04917-f003:**
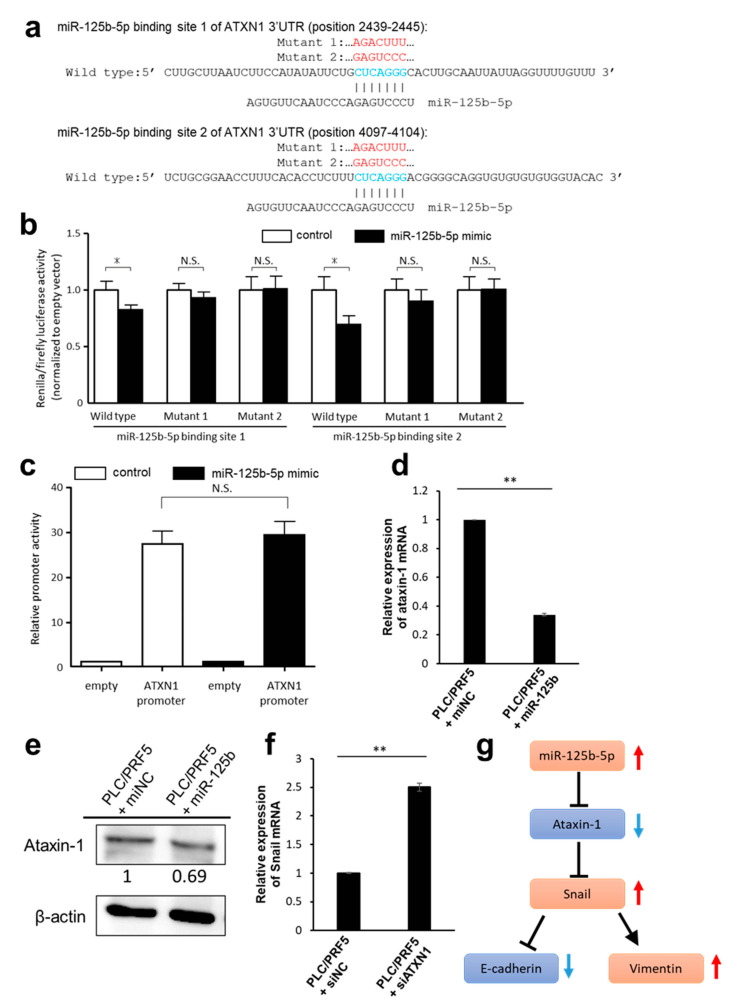
*ATXN1* as a target gene of miR-125b-5p in HCC cells. (**a**) Target Scan predicted two putative binding sites of miR-125b-5p (position 2439-2445 and 4097-4104, shown in blue) in the *ATXN1* 3′-UTR region. The mutated sequences (shown in red) of putative binding sites 1 and 2 are shown as *ATXN1* 3′-UTR Mutant 1 or Mutant 2, respectively. The indicated sequences (Wild type and Mutant 1 and 2) of *ATXN1* 3′-UTR were cloned to psiCHECK2 vector. (**b**) psiCHECK2 vectors containing the wild type or mutant sequences, or the empty vector were co-transfected with miR-125b-5p. Renilla luciferase activity was normalized to firefly luciferase. Fold-change values were normalized to empty vector and mimic control-treated cells. Data are expressed as the mean ± standard deviation (SD; *n* = 4). * Statistically significant difference versus control miRNA treatment (unpaired Student’s *t*-test, *p* < 0.05). (**c**) HEK293T cells were co-transfected with microRNA mimics and luciferase reporter plasmids driven by the promoter fragments of *ATXN1*. Luciferase activities in these cells were measured using the Dual-Luciferase Reporter Assay System. Data are expressed as the mean ± standard deviation (SD; *n* = 4). No significant difference between control and miR-125b-5p mimics treated cells (unpaired Student’s *t*-test). (**d**) The relative expression level of *ATXN1* mRNA in PLC/PRF5-miR125b or PLC/PRF5-miNC was determined by RT-PCR. (**e**) The expression of ATXN1 protein in PLC/PRF5-miNC and PLC/PRF5-miR125b cells was examined by western blot analysis. Densitometry analysis indicates relative protein levels from one representative of three independent experiments. Numbers below the bands represent protein expression normalized to β-actin. (**f**) The relative mRNA levels of *Snail* in PLC/PRF5 cells transfected with siRNA-*ATXN1*(PLC/PRF5-siATXN1) or control siRNA (PLC/PRF5-siNC) were determined by RT-PCR. (**g**) Schematic diagram of miR-125-5p-ATXN1-axis and subsequent induction of EMT. ** *p* < 0.01.

**Figure 4 cancers-13-04917-f004:**
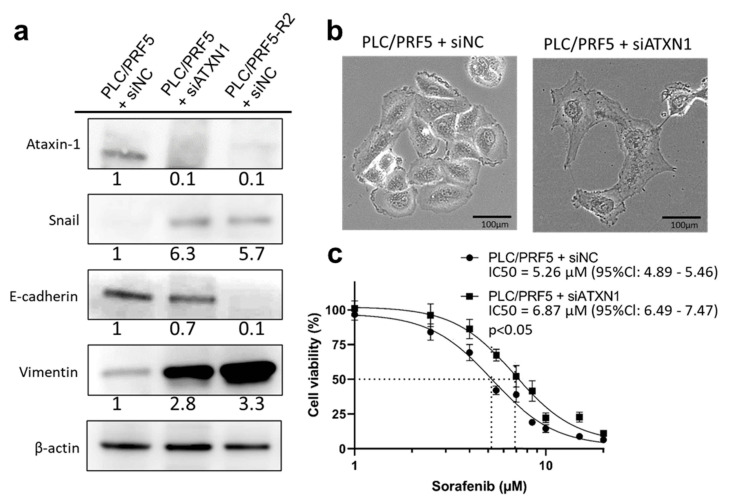
EMT and sorafenib resistance in PLC/PRF5-siATXN1 cells. (**a**) The protein expression of ATXN1 and EMT markers (E-cadherin, Snail, vimentin) in PLC/PRF5-siNC, PLC/PRF5-siATXN1, and PLC/PRF5-R2-siNC cells were examined by western blot analysis. Densitometry analysis indicates relative protein levels from one representative of three independent experiments. Numbers below the bands represent protein expression normalized to β-actin. (**b**) Representative images of PLC/PRF5-siNC and PLC/PRF5-siATXN1 cells under observation with a stereoscopic microscope are shown. Scale bars, 100 μm. (**c**) The viability of PLC/PRF5-siNC and PLC/PRF5-siATXN1 cells treated with various concentrations of sorafenib was determined by WST assay.

**Figure 5 cancers-13-04917-f005:**
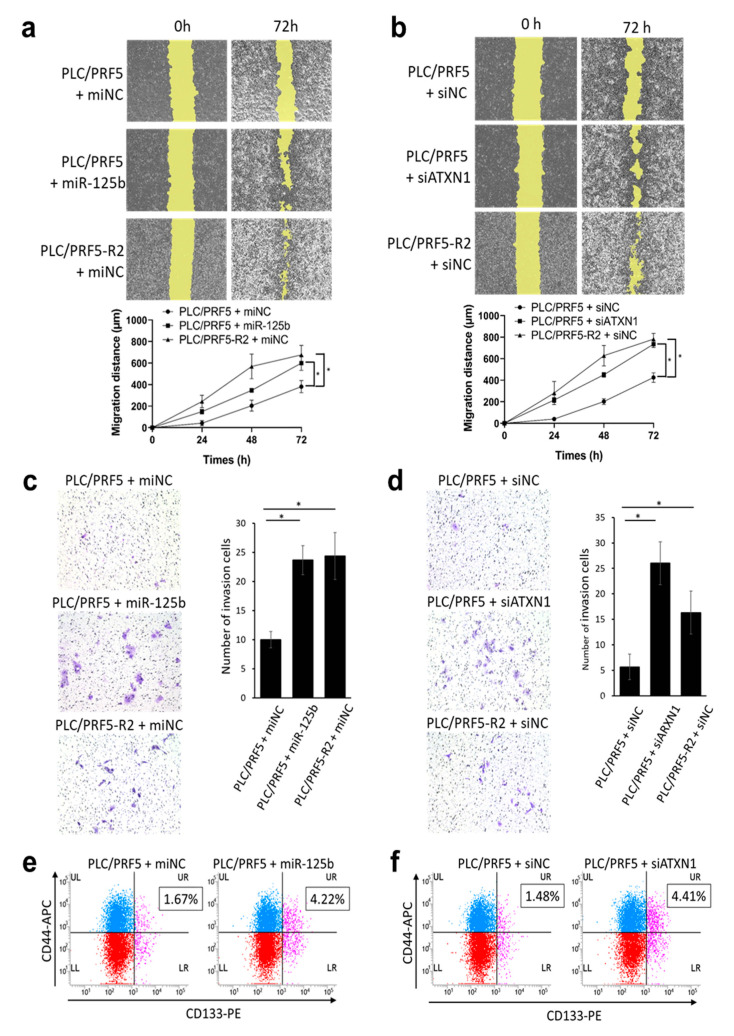
Forced expression of miR-125b-5p or knockdown of *ATXN1* gene increases migration, invasion capability, and cancer stemness of HCC cell lines. (**a**) Representative image of the wound-healing assay in PLC/PRF5-miNC, PLC/PRF5-miR125b, and PLC/PRF5-R2-miNC cells (upper panel). The average migration distance was calculated from the wound healing area, and the time-dependent distance of migration is depicted in the lower panel (mean ± SD; *n* = 3). (**b**) Similarly, representative images of the wound-healing assay and time-dependent distance of migration in PLC/PRF5-siNC, PLC/PRF5-siATXN1, and PLC/PRF5-R2-siNC cells are shown in the lower panel (mean ± SD; *n* = 3). (**c**) Representative images of the invasion assay in PLC/PRF5-miNC, PLC/PRF5-miR125b, and PLC/PRF5-R2-miNC cells (left panel). The number of invading cells was counted, and the average numbers of invading cells are shown (right panel). (**d**) Similarly, representative invasion assay images and quantified invading cells in PLC/PRF5-siNC, PLC/PRF5-siATXN1, and PLC/PRF5-R2-siNC cells are shown. (**e**) Expression of CD44 and CD133 in PLC/PRF5-miNC and PLC/PRF5-miR125b cells was evaluated by two-color flow cytometry using mouse anti-human CD44 monoclonal antibody (APC) and mouse anti-human CD133 monoclonal antibody (PE). The percentage of CD44 + CD133 + cells based on the analysis of 10,000 cells is indicated. (**f**) Expression of CD44 and CD133 in PLC/PRF5-siNC and PLC/PRF5-siATXN1 cells was analyzed by flow cytometry and the percentage of CD44 + CD133 + cells was determined. * *p* < 0.05.

**Figure 6 cancers-13-04917-f006:**
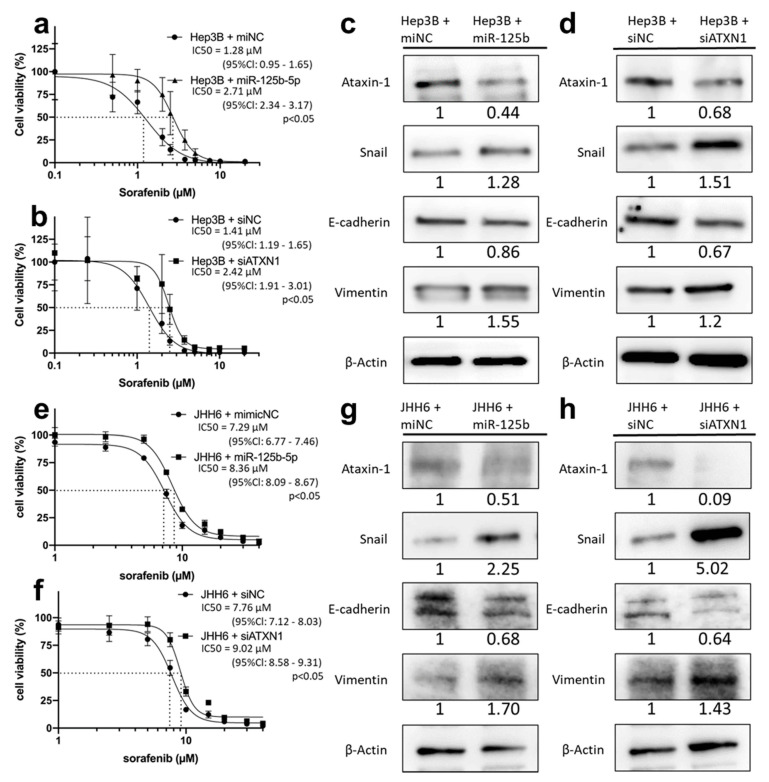
Forced expression of miR-125b-5p or knockdown of *ATXN1* gene induces sorafenib resistance and EMT in Hep3B and JHH6 cells. (**a**) Hep3B cells were transfected with miR-125b-5p mimic (Hep3B-miR125b) or miNC (Hep3B-miNC). They were exposed to sorafenib, and cell viability was assessed by WST assay. (**b**) Hep3B cells were transfected with siATXN1 (Hep3B-siATXN1) or siNC (Hep3B-siNC) and exposed to sorafenib. Cell viability was assessed by WST assay. (**c**) Expression of ATXN1 and EMT markers (E-cadherin, Snail, vimentin) in Hep3B-miNC and Hep3B-miR125b cells was examined by western blot analysis. Densitometry analysis indicates relative protein levels from one representative of three independent experiments. Numbers below the bands represent protein expression normalized to β-actin. (**d**) Expression of ATXN1 and EMT markers in Hep3B-siNC and Hep3B-siATXN1 cells was examined by western blot analysis. (**e**) JHH6 cells were transfected with miR-125b-5p mimic (JHH6-miR125b) or miNC (JHH6-miNC). They were exposed to sorafenib, and cell viability was assessed by WST assay. (**f**) JHH6 cells were transfected with siATXN1 (JHH6-siATXN1) or siNC (JHH6-siNC) and exposed to sorafenib. Cell viability was assessed by WST assay. (**g**) Expression of ATXN1 and EMT markers (E-cadherin, Snail, vimentin) in JHH6-miNC and JHH6-miR125b cells was examined by western blot analysis. (**h**) Expression of ATXN1 and EMT markers in JHH6-siNC and JHH6-siATXN1 cells was examined by western blot analysis.

**Figure 7 cancers-13-04917-f007:**
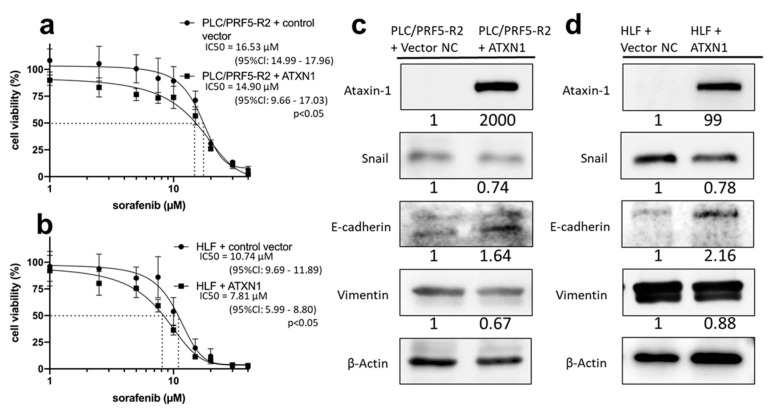
Forced expression of *ATXN1* gene reduces sorafenib resistance and MET in PLC/PRF5-R2 and HLF cells. (**a**,**b**) PLC/PRF5-R2 and HLF cells were transfected with *ATXN1* expressing vector (PLC/PRF5-R2-ATXN1, HLF-ATXN1) or the control vector (PLC/PRF5-R2-Vector NC, HLF-Vector NC). They were exposed to sorafenib, and cell viability was assessed by WST assay. (**c**,**d**) Expression of ATXN1 and EMT markers (E-cadherin, Snail, vimentin) were examined by western blot analysis. Densitometry analysis indicates relative protein levels from one representative of three independent experiments. Numbers below the bands represent protein expression normalized to β-actin.

**Figure 8 cancers-13-04917-f008:**
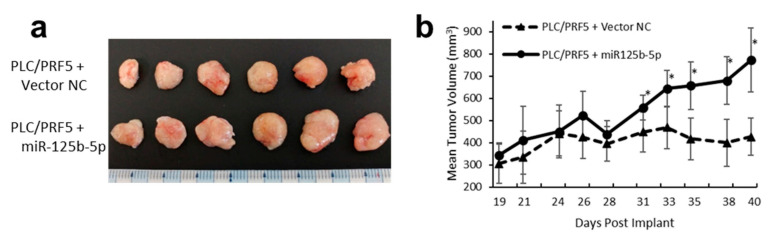
PLC/PRF/5 cells treated with miR-125b-5p expressing vector or negative control (NC) vector were implanted s.c. in the flank of mice (5 × 10^6^ per animal). Treatment was initiated on day 19 when all tumors reached 200 mm^3^ in size. Sorafenib or vehicle control was administered orally, once a day, for 21 days at a dose of 30 mg/kg. There was no lethality in any group. (**a**) Images of the tumors from all mice in each group (*n* = 6). (**b**) Tumor dimensions were recorded three times a week starting with the first day of treatment (*n* = 6). Data represent mean ± SD. * *p* < 0.05 compared with the control group.

**Figure 9 cancers-13-04917-f009:**
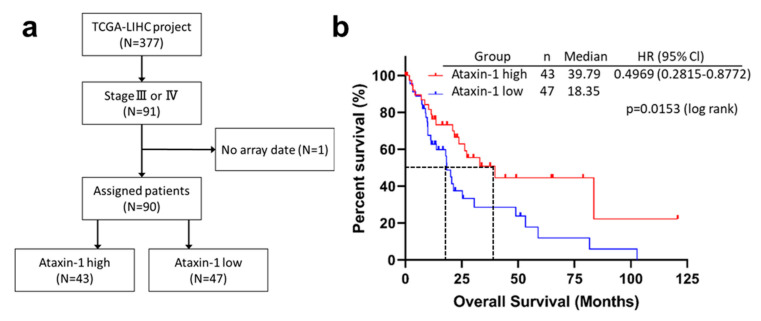
ATXN1 expression was significantly associated with the overall survival of patients with advanced HCC. (**a**) A flowchart of cohort selection from the public database. Among the TCGA–LIHC dataset, patients with Stage III or IV (*n* = 91) were selected, and one patient was excluded due to lack of array data. Ninety patients were divided into two groups according to high or low expression of ATXN1. (**b**) Kaplan–Meier survival curves for overall survival. HR, hazard ratio; CI, confidence interval. The *p*-value was calculated by log-rank test.

## Data Availability

Publicly available datasets were analyzed in this study. This data can be found here: [https://portal.gdc.cancer.gov/projects/TCGA-LIHC] (accessed on 9 February 2021).

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
