# Peer review of "MiR-125b-5p Is Involved in Sorafenib Resistance through Ataxin-1-Mediated Epithelial-Mesenchymal Transition in Hepatocellular Carcinoma"

_cancers, 2021, doi:10.3390/cancers13194917_

Round 1
Reviewer 1 Report
No further comments.
Reviewer 2 Report
The authors have addressed this Reviewer main comments and concerns making this manuscript suitable for publication.
This manuscript is a resubmission of an earlier submission. The following is a list of the peer review reports and author responses from that submission.
Round 1
Reviewer 1 Report
The authors have identified upregulation of miR-125-5p as a novel microRNA underlying sorafenib resistance in 2 sorafenib-resistant HCC cell lines derived from the PLC/PRF5 cell line. Further mechanistic studies have identified that this is due to inhibition of the ataxin-1 target gene which in turn induces EMT and sorafenib resistance.
The mechanistic work is comprehensive and very well presented.
Major points:
The major limitation of this work is whether the findings are generalizable. The authors have validated their findings in only one other cell line - the Hep3B cell line. Indeed, it is notable that both the PLC/PRF5 and the Hep3B cell lines are Hep B positive cell lines, so its difficult to know if this finding may be a hepatitis B related phenomena or generalizable to all HCC. Validation in other cell lines is needed to clarify this. Certainly clinically this is important as patients with HCC and underlying Hep B are known to respond less well to Sorafenib compared to HCC due to other aetiologies, especially Hepatitis C.
Minor points:
In the second paragraph of the introduction the discussion regarding serial use of sorafenib and lenvatinib and vice versa is not clinically correct. Sorafenib and Lenvatinib have both demonstrated equivalent efficacy in the first line setting but there is no level I evidence to support sequencing of these two drugs and this approach is contrary to current Clinical Practice Guidelines for management of HCC. eg ESMO guidelines.
In the final paragraph of the discussion the authors write :
".. by comparing the expression status of 505 miR-125b-5p and the effect of sorafenib in patients in the clinical setting, it is expected that 506 miR-125b-5p will be established as an effective drug selection marker for treatment selection in patients with HCC."
There is no certainty that this will be the case and I would recommend the authors temper this line along the lines that this finding needs to be validated clinically - ideally in prospective trials but certainly in patient samples to see if this finding can be validated in real life and is not just a cell line phenomena.
Reviewer 2 Report
In this report, Hirao and collaborators identify miR-125b-5p as a determinant of sorafenib resistance in hepatocellular carcinoma (HCC) cells. Mechanistically, the authors propose that miR-125b-5p mediates the induction of epithelial-mesenchymal transition (EMT) in HCC cells, and it is known that tumor cells with a mesenchymal phenotype are more aggressive and chemoresistant. Ataxin-1 (ATXN-1) is demonstrated to be a target of miR-125b-5p, and HCC cells resistant to sorafenib have a lower expression of ATXN-1. Manipulation of miR-125b-5p and ATXN-1 levels in HCC cells change their migratory and invasive properties and sorafenib sensitivity, along with the expression of cancer stem cell markers. Finally, the authors find an association between higher ATXN-1 expression and enhanced HCC patients’ survival. This is an interesting study with novel and relevant findings. Nevertheless, I have some aspects for the authors to consider that may increase the robustness of their findings.
- ATXN-1 should be better described in the Introduction, along with the most relevant references on this gene and its functions in the cancer literature.
- It would be relevant to evaluate the effects of ATXN-1 overexpression on the phenotype, E-cadherin and Snail expression, and responses to sorafenib in PLC/PRF5-R cells. The authors could test this by transfecting these cells with an ATXN-1 expression construct resistant to miR-125b-5p.
- There is little known about ATXN-1 in cancer, and less so in HCC. To further support the role of this gene in the preservation of the epithelial phenotype in liver cells it would be interesting to examine the effects of ATXN-1 expression in HCC cell lines that have a more mesenchymal phenotype, such as SNU-449 cells. Could ATXN-1 reverse their constitutive mesenchymal phenotype?
- Figs. 2, 3,4, and 6 present western blot analyses of different proteins. It would be important to indicate how many times these experiments were performed. The authors should provide statistical analyses of the quantification of these independent experiments.
Minor points.
- miR-125b-5p should be consistently named throughout the manuscript. Some times the authors write just miR-125b.
- In the legend to Fig. 1 it reads “Differentially expressed genes in…”, it should be “Differentially expressed miRNAs in…”.
Reviewer 3 Report
The manuscript by Hirao et al. shows that mir-125b-5p exerts its effects through ataxin-1-mediated EMT in HCC.
Even though the study is novel, the major point is that fundamental experiments are missing, such as the luciferase assay activity, to definitely show that mir-125b-5p exerts its effects by increasing the transcriptional activity of the Ataxin-1 promoter. It would also be relevant to show the effects of miR-125b-5p in at least an in vivo xenograft mouse model. Other important point that is not completely clear is why ataxin was chosen amongst the top 50 regulated genes. It is important to show the list of these genes.
Other minor points include:
The Introduction should include the reported studies already existent on miR-125-5p in HCC and perhaps reduce the comparison between sorafenib and Lenvatinib, that is out of the scope of this manuscript.
The protocol for wound healing assay is not very clear.
Other cell viability assay should be present to compare the effects of sorafenib-induced cell death. Maybe FACS analyses could be performed.
It was very surprising to find results that are described, and figures of results shown in the discussion (Figure S4 and S5). This should be rewritten.